# Quality Characteristics of Senior-Friendly Gelatin Gels Formulated with Hot Water Extract from Red Maple Leaf as a Novel Anthocyanin Source

**DOI:** 10.3390/foods10123074

**Published:** 2021-12-10

**Authors:** Dong-Heon Song, Tae-Wan Gu, Hyun-Wook Kim

**Affiliations:** Department of Animal Science & Biotechnology, Gyeongsang National University, Jinju 52725, Korea; timesoul@naver.com (D.-H.S.); ktw4653@naver.com (T.-W.G.)

**Keywords:** antioxidant, anthocyanin, colorant, gelatin gel, jelly food, red maple leaf

## Abstract

The objectives of this study were to evaluate antioxidant capacity of hot water extract from red maple leaf with different extraction times (experiment I) and to determine their impacts on color, free anthocyanin content, and hardness of gelatin gels (experiment II). In experiment I, hot water extraction time (30, 60, 120, 180, and 360 min at 60 °C) was fixed as a main effect. The different extraction times had no impacts on total polyphenol content and DPPH radical scavenging activity (*p* > 0.05). However, extraction time for 360 min could decrease anthocyanin content as well as ferric reducing antioxidant power (*p* < 0.05). In experiment II, 6%, 18%, and 30% gelatin gels were prepared without/with red maple leaf extract (1000 mg/L). The red maple leaf extract significantly increased redness, yellowness, and hardness, but decreased free anthocyanin content. Such impacts were obviously observed at high gelatin concentration. Thus, red maple leaf extract could be a novel anthocyanin source for improving antioxidant capacity and reddish color of gelatin gels. However, the addition amount of red maple leaf extract may be limited in the development of senior-friendly jelly food for soft texture in that it could increase the hardness of the gelatin gel.

## 1. Introduction

Aging in the human body causes malfunction of physiological activity, which in turn promotes the production of excessive free radicals, particularly reactive oxygen species [1,2]. The accumulation of free radicals can induce oxidative stress, which has been pointed out as a major cause of accelerated arteriosclerosis, arthritis, and dementia [3,4]. In this regard, the protective effects of food antioxidants against oxidative damages, through the increase of antioxidant capacity in blood and the removal of free radicals generated, has been extensively studied [5,6,7]. Not only plants in general, but also seeds, contain various antioxidative compounds, mainly polyphenols including flavonoids, phenolic acids, lignans, essential oils, and alkaloids [7]. Anthocyanins present in red, purple, and blue plants are water-soluble flavonoid compounds with excellent antioxidant activity [5,7]. Black beans, blueberries, and purple carrots are abundant in anthocyanins and have been reported to have not only antioxidant activity but also anti-cancer, anti-inflammatory, anti-allergic, anti-viral, and immune-enhancing functions [8,9,10,11]. Thus, the plant resources containing high number of anthocyanins are considered as a healthy food ingredient for improving oxidative damages.

Maple (*Acer palmatum* Thunb.) is naturally grown in East Asia, North America, and Europe, and 20% of its total compounds are composed of flavonoids [12]. In Korea, where the temperature changes in the four seasons are obvious, maple leaves turn red color in autumn when the temperature is relatively low (below about 5 °C of daily minimum temperature). In addition, the temperature change can affect not only the foliar color change but also the various physiological functions [13]. It has been known that the color change in maple leaf is related to the exposure of anthocyanins following chlorophylls destruction, and red maple leaves contain about 4 times more anthocyanins than green leaves [14]. In fact, strong antioxidant effects of red maple leaf have been reported [12,13,14], but its practical utilization as a food ingredient is still limited. In particular, there has been little to no research on establishment of pretreatment method (e.g., extraction condition) of antioxidants from red maple leaf and its application strategy as a dietary antioxidant.

In developed countries, the proportion of elderly (over 65 years old) has been rapidly increasing [15], and the development of senior-friendly food considering the dietary characteristics of the elderly is recognized as one of the main fields in the food industry. Texture and nutritional value are considered as the most important factors in the development of senior-friendly foods. Jelly food, which is easy to chew, is one of the types that is extensively used as a senior-friendly food [16], since it has the advantage of controlling texture and mixing with other food materials. Moreover, utilization of bioactive substances such as antioxidants have been studied to enhance the nutritional and functional properties of senior-friendly jelly foods [17,18].

In general, addition of antioxidants to processed foods is effective not only in improving oxidation stability of food, but also in scavenging free radicals from the body of those who consume the foods [8]. In particular, for elderly people, the fact that intake of antioxidant foods can improve blood lipid concentration and immune function may be important for maintaining a healthy life [19]. Thus, it could expected that use of red maple leaf extract that contains abundant anthocyanin in the manufacture of senior-friendly jelly food can have a positive contribution on the health of the elderly. However, there has been little to no literature on optimal condition of anthocyanin extract from red maple leaf and its impacts on general quality characteristics (color and texture) of jelly foods. Therefore, the objectives of this study were to 1) evaluate the antioxidant properties of red maple leaf extract with different time for hot water extraction, and 2) determine the quality characteristics of a senior-friendly gelatin gel formulated with red maple leaf extract as a novel ingredient of antioxidant and red colorant.

## 2. Materials and Methods

### 2.1. Experimental Design and Arrangement

To preferentially determine the effect of hot water extraction time on antioxidant capacity of red maple leaf extract (experiment I), an experimental design was set as completely randomized block design with three independent batches, where extraction time (30, 60, 120, 180, and 360 min at 60 °C) was fixed as a main effect. In experiment II, to evaluate the effect of hot-water extract of red maple leaf on physicochemical properties of senior-friendly gelatin gels, an experimental arrangement was a 2 (with or without red maple leaf extract) × 3 (gelatin concentration) factorial set in a completely randomized block design with three independent batches, where red maple leaf extract effect, gelatin concentration effect, and their interaction effect were fixed as main effects.

### 2.2. Hot Water Extraction of Red Maple Leaf (Experiment I)

The red maple leaves used in this study were collected from maple trees (*Acer palmatum* Thunb.) that grow wild in Jinju, Gyeongsangnam-do in November 2019. The collected red maple leaves were washed several times using distilled water, dried in a 50 °C dry-oven (OF-22 GW, Jelo Tech., DaeJeon, Korea) for 3 h, and pulverized. The averaged drying yield of red maple leaf was 0.45 ± 0.03%. The red maple leaf powder was mixed with 40 volumes of distilled water, and the hot water extraction of the sample was performed at 60 °C for 30, 60, 120, 180, and 360 min. Previously, Jun et al. [20] suggested that the optimal condition of blueberry extract containing anthocyanins using hot water was at 60 °C for 3 h, based on the result, the extraction temperature and times were set. After the hot-water extraction, the extract was centrifuged at 1000× *g* for 10 min (4 °C), and the supernatant was filtered through a filter paper (Whatman No. 4). The filtrate was freeze-dried using a freeze dryer (PVTFD10R, Ilshin biobase, Daejeon, Korea), pulverized, vacuum-packaged in a PE/Nylon bag, and stored in a refrigerator until further analysis.

### 2.3. Analysis of Antioxidant Capacity of Red Maple Leaf Extract

#### 2.3.1. Total Phenol Content

Total phenol contents of red maple leaf were determined in triplicate by the Folin–Ciocalteu method [21]. A total 100 mg of freeze-dried red maple leaf powder were dissolved in 1 mL distilled water. One hundred microliters of the mixture were mixed with 2.8 mL distilled-deionized water, two milliliters of 2% sodium carbonate (Na_2_CO_3_), and 0.1 mL of 50% Folin-Ciacalteau reagent. The mixture was incubated at room temperature for 30 min, and the absorbance of the mixture was read at 750 nm. Gallic acid (0–1000 μg/mL) was used as a standard material, and the total phenol content was expressed as mg gallic acid equivalents per g of lyophilized sample (GAE/g).

#### 2.3.2. Anthocyanin Content

Anthocyanin content of freeze-dried red maple leaf powder was determined in triplicate according to the method of Giusti and Wrolstad [22]. The freeze-dried sample dissolved in distilled water was mixed with 25 mM potassium chloride buffer (pH 1.0) and 0.4 M sodium acetate buffer (pH 4.5) for 15 min, respectively, and the absorbance of mixtures were measured at 510 nm and 700 nm. Anthocyanin content was calculated using following equation and expressed as mg per 100 g of lyophilized sample.
Anthocyanin content (mg/100 g) = (A × MW × DF × 1000)/(ε × 1)
where A is (Abs_510nm_ − Abs_700nm_) of pH 1.0 sample—(Abs_510nm_ − Abs_700nm_) of pH 4.5 sample, MW is the molecular weight of cyanidin-3-glucoside (484.83 g/mol), DF is the dilution factor, and ε is the molar absorptivity (26,900 L cm^−1^ mol^−1^).

#### 2.3.3. DPPH Radical Scavenging Activity

1,1-Diphenyl-2-picryl-hydrazyl (DPPH) radical scavenging ability of red maple leaf was determined in triplicate according to the method of Sharma and Bhat [23]. One hundred microliters of diluted samples (0.05, 0.1, 0.2, 0.4, and 0.8 mg/mL) was mixed with 4 mL of methanol and 0.1 mL of 200 μM DPPH reagent and reacted in a dark room for 30 min. Absorbance of the mixture was read at 517 nm, and DPPH radical scavenging activity was expressed as a percentage difference in the absorbance at 517 nm between control and samples.

#### 2.3.4. Ferric Reducing Antioxidant Power

Ferric reducing ability of red maple leaf was determined in triplicate according to the ferric reducing antioxidant power (FRAP) assay of Othman et al. [24]. FRAP reagent was prepared by mixing 0.3 M acetate buffer (pH 3.6), 20 mM ferric trichloride hexahydrate (FeCl_3_·6H_2_O), and 10 mM 2,4,6-tripyridyl-s-triazine (TPTZ) at 10:1:1 ratio. One hundred microliters of diluted samples (0.05, 0.1, 0.2, 0.4, and 0.8 mg/mL) was reacted with 0.3 mL of distilled water and 3 mL of FRAP reagent for 4 min, and the absorbance of the mixture was read at 593 nm. Ferrous sulfate heptahydrate (FeSO_4_·7H_2_O) was used as a standard material, and FRAP was expressed as FRAP μmoles per liter (FRAP μmoles/L).

### 2.4. Manufacture of Gelatin Gels with Red Maple Leaf Extract (Experiment II)

Six treatments of gelatin gels without/with red maple leaf extract at three different protein concentrations were prepared. The commercial pig skin gelatin (91.12 g/100 g of crude protein, 200 bloom) was dissolved in distilled water to produce 6%, 18%, and 30% (*w*/*v*) of protein concentration in gelatin gels, in which the protein concentrations were set considering the target hardness for teeth (50,000–500,000 N/m^2^), gum (20,000–50,000 N/m^2^), and tongue intakes (<20,000 N/m^2^), respectively [25]. Red maple leaf extract was prepared through 120 min of hot water extraction, and 1000 mg/L of the extract was mixed each gelatin solution at 40 °C. The mixture was gelated at room temperature for 2 h and used for further analysis.

### 2.5. Physicochemical Analysis of Gelatin Gels with Red Maple Leaf Extract

#### 2.5.1. Surface Color Measurement

Surface color of gelatin gels with red maple leaf extract was measured using a colorimeter (CR-400, Minolta, Osaka, Japan) equipped with an 8 mm diameter observer. According to the manufacturer’s manual, the instrument was calibrated using a calibration tile (CIE L*: +93.01, CIE a*: −0.25, CIE b*: +3.50) under a D_65_ illumination source. CIE L*, a*, and b* values were recorded from six random locations.

#### 2.5.2. Free Anthocyanin Content

To determine the amount of free anthocyanin released from gelatin gels, one gram of sample was hydrolyzed in 12 mL of 1% HCl at 4 °C for 48 h. The hydrolysate was centrifuged at 4000× *g* for 15 min (4 °C), and the supernatant used to determine anthocyanin content according to the method described in 2. 3. 2. The free anthocyanin content was expressed as mg per 100 g of gelatin gels.

#### 2.5.3. Hardness

Hardness of gelatin gels was determined according to the method of senior-friendly foods in Korean industrial standard [25]. The sample was cut into 10 × 10 × 10 mm^3^ and hardness was measured using a texture analyzer (CT3, Brookfield Engineering Laboratories, Inc., Middleboro, MA, USA). The sample was placed in the center of the plate, and the hardness curve was taken from twice 70% compression with 1 cm diameter probe. Analysis conditions were set as pre-test speed 1 mm/s, test speed 2 mm/s, post-test speed 10 mm/s, and strain 70% compression.

### 2.6. Statistical Analysis

Analysis of variance was performed on all variables measured using the general linear model (GLM) procedure in SPSS 18.0 program (SPSS Inc., Chicago, IL, USA). For the analysis of antioxidant activity of maple leaf extract, the extraction time was considered as a main effect. In the case of physicochemical analysis of gelatin gels, protein concentration, the addition of maple leaf extract, and their interaction were fixed as main effects. Duncan’s multiple range test was used to determine the significance of the differences between treatments (*p* < 0.05).

## 3. Results and Discussion

### 3.1. Total Phenol and Anthocyanin Contents of Hot Water Extract from Red Maple Leaf

Total phenol and anthocyanin contents of red maple leaf extract with different hot water extraction for 30, 60, 120, 180, and 360 min are shown in Figure 1. Red maple leaf extracts showed similar total phenol contents, regardless of different extraction times (*p* > 0.05). This could indicate even the shortest time condition of 30 min might be sufficient to obtain extractable polyphenols from red maple leaf. As similar results, Choi et al. [26] found that polyphenol content of hot water extract from microalgae was not changed after 2.5 min at 60 °C. Piljac-Žegarac et al. [27] reported that 30 min of hot water extraction time resulted in the highest total polyphenol content of blueberry leaf extract. An increase in extraction time significantly affected anthocyanin content in the red maple extract, in particular, hot water extraction for 360 min obviously resulted in a decrease in anthocyanin content of red maple extract (*p* < 0.05). According to Schmitzer et al. [28], in the leaves of *Acer* cultivars, cyaniding-3-glucoside is known as one of the most abundant anthocyanins.

In general, antioxidant capacity of plant extract could be greatly related to the extracted amount of phenolic compounds, especially polyphenols including phenolic acids, flavonoids, stilbenes, and lignans regarded as polyphenols [29]. Flavonoids, which account for approximately half of identified polyphenolic compounds, include flavanones, isoflavones, and anthocyanins [7]. In particular, anthocyanin contained in plant resources, a water-soluble compound, has been reported to have a strong antioxidant capacity in human body, and berries, beetroot, pomegranate, and black bean are an excellent food abundant in dietary anthocyanin [30]. In many previous studies, thus, development and optimization of extraction methods to obtain as much polyphenols as possible from the selected plant source were extensively studied. According to Silva et al. [31], solvent and temperature are critical parameters affecting the yield of anthocyanin extract and its antioxidant activity. Moreover, it has been reported that an increase of extraction time could also decrease anthocyanin content in blackberry extract [32]. Thus, to guarantee high antioxidant capacity of red maple leaf extract, our results show that optimal time of hot water extraction at 60 °C could be within 180 min, since the extraction condition has little to no impacts on polyphenol as well as anthocyanin contents.

### 3.2. Antioxidant Capacity of Hot Water Extract from Red Maple Leaf

Antioxidant capacity of red maple leaf extract was determined through DPPH radical scavenging activity (Figure 2) and ferric reducing antioxidant powder (FRAP) assay (Figure 3). DPPH radical scavenging activity of red maple leaf extracts showed concentration-dependent manner (Figure 2a), but no significant interaction between extraction time and concentration was observed. As a result, the red maple leaf extracts presented similar 50% radical scavenging activity (IC_50_) value as follows (*p* > 0.05); 30 min (611 ppm), 60 min (603 ppm), 120 min (599 ppm), 180 min (576 ppm), and 360 min (617 ppm). Ferric reducing antioxidant power (FRAP) of red maple leaf extracts with different extraction time is shown in Figure 3. An increase in extraction time by 120 min increased FRAP value of red maple leaf extract, but the FRAP value decreased after 180 min of extraction time. The extraction time of 120 min caused significantly higher FRAP value compared to extraction for 30 min or 360 min (Figure 3b). In general, the amount of polyphenolic compounds is highly correlated with antioxidant capacity, thus, many previous studies have found that extraction of a large amount of polyphenol can be expected to produce an extract with excellent antioxidant activity. In this study, hot water extraction for 360 min had no impact on polyphenol content as well as DPPH radical scavenging activity of red maple leaf extract. However, the decreased FRAP value over 120 min might be associated with partial thermal-degradation of heat-sensitive polyphenolic compounds including anthocyanins. Considering the antioxidant content and activity contained in the maple leaf extract, thus, the most appropriate hot water extraction time at 60 °C may be 120 min.

### 3.3. Color Characteristics of Gelatin Gels with Red Maple Leaf Extract

Exterior photos of 6%, 18%, and 30% gelatin gels formulated with 1000 mg/L of red maple leaf extract are shown in Figure 4. At 6% concentration (Figure 4a), the gelatin gel showed typical translucent phase, but the addition of 1000 mg/L of red maple leaf extract caused a distinct orange color (Figure 4d). An increase in gelatin concentration caused the formation of a yellowish gel visually, but between the extract treatments (Figure 4d–f), no difference in the color properties was observed with the naked eye regardless of gelatin concentration.

Instrumental color characteristics of 6%, 18%, and 30% gelatin gels formulated with 1000 mg/L of red maple leaf extract are shown in Table 1. In all color parameters evaluated in this study, including CIE L* (lightness), CIE a* (redness), CIE b* (yellowness), and hue angle, significant interaction between gelatin gel concentration and red maple leaf extraction addition was found. Lightness of gelatin gels decreased with increasing gelatin concentration, and addition of red maple leaf extract also decreased the lightness of gelatin gels (*p* < 0.001). The lowest lightness was observed for 18% and 30% gelatin gels with red maple leaf extract (*p* < 0.05). An increase in gelatin concentration and the addition of red maple leaf extract increased redness and yellowness of gelatin gels (*p* < 0.001). In particular, 30% gelatin gels with red maple leaf extract showed the highest redness (*p* < 0.05). According to Chung et al. [33], color stability of anthocyanins can be improved by interaction with bio-polymers. In this study, thus, the increased gelatin concentration may contribute to the improvement of color stability of anthocyanins through an interaction of gelatin-anthocyanin molecules. In terms of hue angle, the addition of red maple leaf extraction significantly affected the hue angle of gelatin gels. At the same gelatin concentration, treatment gel groups with red maple leaf extraction had significantly lower hue angle than the gels without the extract. Moreover, the lowest hue angle, indicating more reddish color, was observed for 18% and 30% gelatin gels with red maple leaf extract. Previously, it has been well-documented that color-change of anthocyanin occurs depending on pH, ranging from red, violet, blue, to green [30]. Commercial pork skin gelatin, which is generally obtained through acid swelling to extract gelatin type A, has mild acidic pH around 4.0–5.0 [25]. Thus, pork skin gelatin based gel could be an excellent bio-polymer to produce reddish color of jelly foods containing anthocyanins. Moreover, our results show that high gelatin concentration might result in the enhancement and stabilization of reddish color of gelatin gels with anthocyanins.

### 3.4. Free Anthocyanin Content of Gelatin Gels with Red Maple Leaf Extract

Free anthocyanin content of 6%, 18%, and 30% gelatin gels formulated with 1000 mg/L of red maple leaf extract is shown in Figure 5. Free anthocyanin content of gelatin gels was significantly affected by gelatin concentration. The 18% and 30% gelatin gels showed lower free anthocyanin content compared to 6% gelatin gel (*p* < 0.05), although the same amount of red maple leaf extract was added. Free anthocyanin content refers to the amount of anthocyanins that can be easily released during digestion, thus, low release amount can indicate the minimal level of diffusive release of anthocyanins [34]. Previously, Betz and Kulozik [34] have found that an increase in protein level (15%, 20%, and 25%) is why protein gels with bilberry extract did not affect the release rate of anthocyanins. When considering the fact that gelatin can form more intensive gel structure compared to other food proteins at the same protein level [25], our results in this current study show that an increase in gelatin concentration can delay the release of anthocyanins, and suggest that this phenomenon might be associated with a barrier effect of gelatin structure and/or an interaction between the gelatin and anthocyanin molecules.

### 3.5. Hardness of Gelatin Gels with Red Maple Leaf Extract

Hardness of 6%, 18%, and 30% gelatin gels formulated with 1000 mg/L of red maple leaf extract is shown in Figure 6. There was a significant interaction between gelatin concentration and extract addition on hardness of gelatin gels. As expected, an increase in gelatin concentration significantly increased the hardness of gelatin gels. Moreover, at the same gelatin concentration, the addition of red maple leaf extract considerably increased the hardness of gelatin gels (*p* < 0.05). Similarly, Tongmai et al. [35] have reported that addition of 13–26% Thai mulberry increased the hardness of gelatin gel. Hwang and Thi [36] have also found that an increase in aronia extract increased the hardness of gelatin gels, and suggest that the pH change due to organic acid contained in aronia extract and/or saccharides could increase the hardness of gelatin gels. As well, it has been suggested that saccharides in aronia extract could be gelated through thermal processing, which would be one of reasons for the increased hardness of gelatin gels with aronia extract [33]. In this study, the pH of red maple leaf extract was around 3.0, which might contribute to the increased hardness of gelatin gels. Moreover, red maple leaves contain about 8% carbohydrates, of which sugar accounts for about 2% [25]. Thus, the acidic pH and saccharides in red maple leaf extract might be major factor increase the hardness of gelatin gels, which were also agreement with the suggestion of Hwang and Thi [36].

Recently, the development of senior-friendly foods has received great attention, mainly in developed countries where the population has been aging, such as Japan and Korea [25]. Gelatin-based jelly food is easy to adjust hardness, thus it is useful in the development of senior-friendly products [25]. In the case of Korea, the senior-friendly food certification program has been applied based on the Korean industrial standards (KS H 4897) [37]. In the certification program, quality standards of senior-friendly food are classified by properties, hardness, viscosity, and nutritional components. In particular, the hardness is sorted into three grades; 1st grade for teeth intake (500,000–50,000 N/m^2^), 2nd grade for gum intake (50,000–20,000 N/m^2^), and 3rd grade for tongue intake (below 20,000 N/m^2^). Based on our results, in aspect to hardness, below 18% gelatin concentration should be applied to produce senior-friendly jelly food. However, since hardness of jelly food can be modified depending on characteristics of ingredients added together (pH, chemical composition, etc.), formulation of jelly food should be designed to reach the target hardness with considering the enhancement impact of hardness due to red maple leaf extract.

## 4. Conclusions

Hot water extraction during 360 min had no impacts on total polyphenol content of red maple leaf extract, but the extraction time over 180 min could decrease anthocyanin content as well as antioxidant capacity of the extracts. Thus, the results showed that 120 min of hot water extraction time at 60 °C may be sufficient to obtain antioxidative compounds from red maple leaf, including anthocyanins. The addition of 1000 mg/L of red maple leaf extract increased redness and yellowness of 6%, 18%, and 30% gelatin gels. Moreover, high gelatin concentration could decrease the free anthocyanin content. Thus, red maple leaf extract could be a novel anthocyanin source for improving antioxidant capacity and reddish color of high protein gelatin gels. However, the addition amount of red maple leaf extract may be optimized for the development of senior-friendly jelly food for soft texture in that it could increase the hardness of the gelatin gel. Furthermore, in order to evaluate the effect of the hot water extract of maple leaf on the functionality of the senior-friendly gels as an anthocyanin source, further studies profiling the anthocyanin composition of red maple leaf extract as well as developing an efficient extraction should be warranted.

## Figures and Tables

**Figure 1 foods-10-03074-f001:**
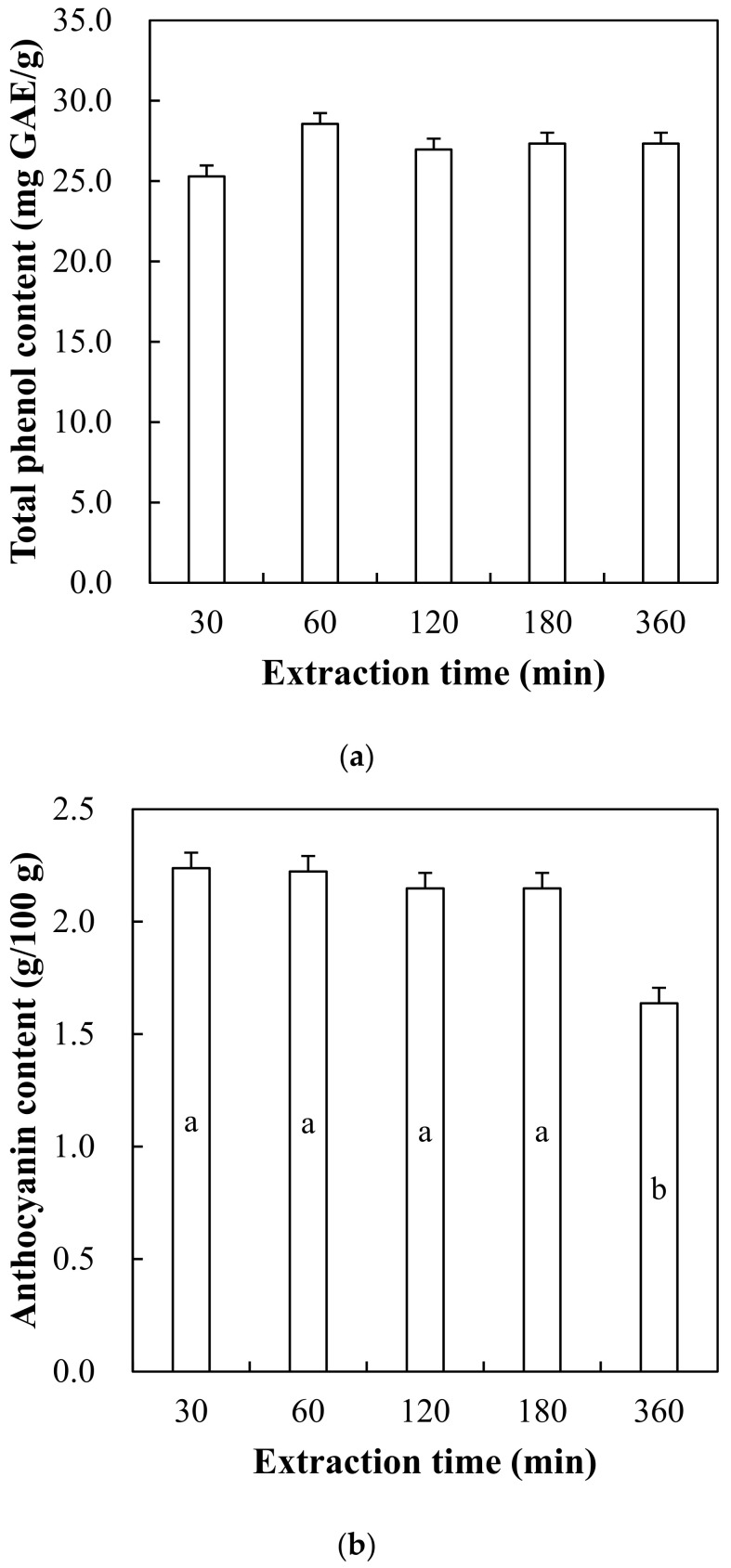
Total phenol content (**a**) and anthocyanin content (**b**) of red maple leaf extracts through hot water extraction (60 °C) for 30, 60, 120, 180, and 360 min. Error bars represent standard error of the means. a,b Means with the same letter are not significantly different (*p* ≥ 0.05).

**Figure 2 foods-10-03074-f002:**
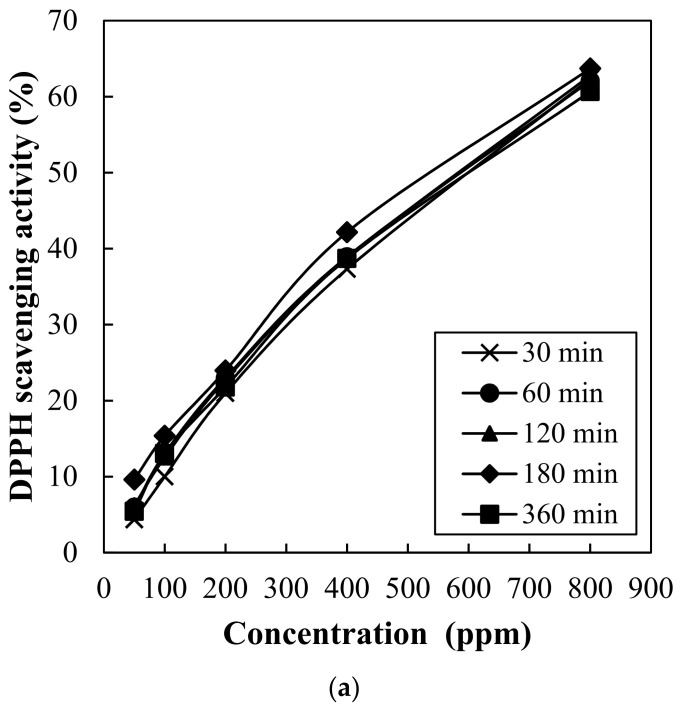
Change in DPPH radical scavenging activity at different concentrations (**a**) and IC_50_ values for DPPH radicals (**b**) of red maple leaf extracts through hot water extraction (60 °C) for 30, 60, 120, 180, and 360 min. AA, L-ascorbic acid (as control). Error bars represent standard error of the means.

**Figure 3 foods-10-03074-f003:**
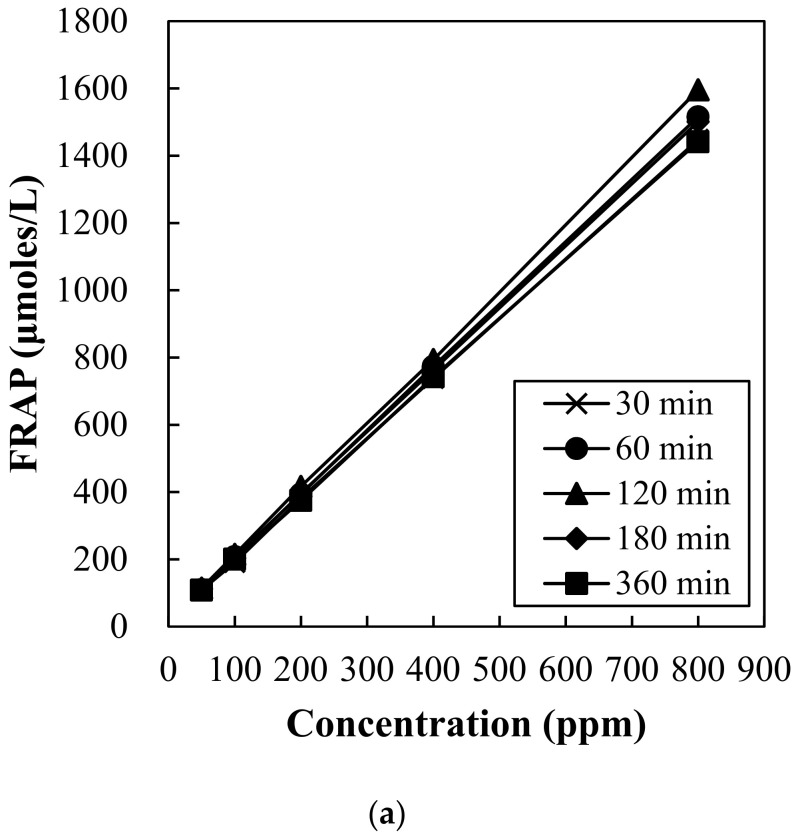
Change in ferric reducing antioxidant power (FRAP) of red maple leaf extracts at different concentration (**a**) and the impact of extraction time on the FRAP value (**b**). Error bars represent standard error of the means. a,b Means with the same letter are not significantly different (*p* ≥ 0.05).

**Figure 4 foods-10-03074-f004:**
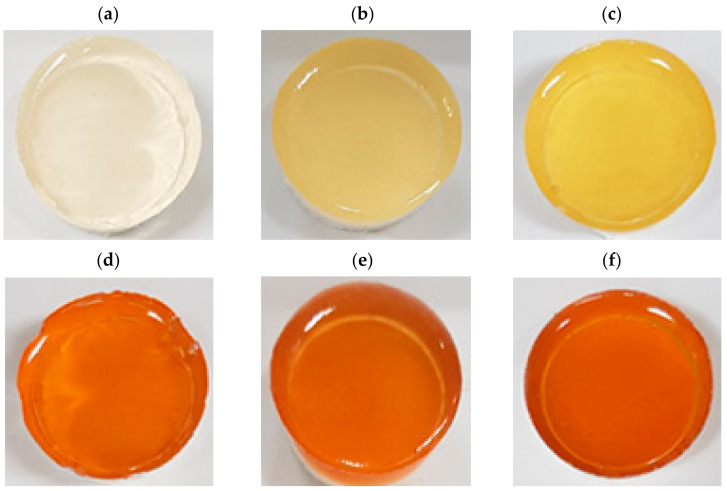
Exterior photos of 6%, 18%, and 30% gelatin gels with or without red maple leaf extract (1000 mg/L). (**a**) 6% gelatin gel; (**b**) 18% gelatin gel; (**c**) 30% gelatin gel; (**d**) 6% gelatin gel with red maple leaf extract; (**e**) 18% gelatin gel with red maple leaf extract; (**f**) 30% gelatin gel with red maple leaf extract.

**Figure 5 foods-10-03074-f005:**
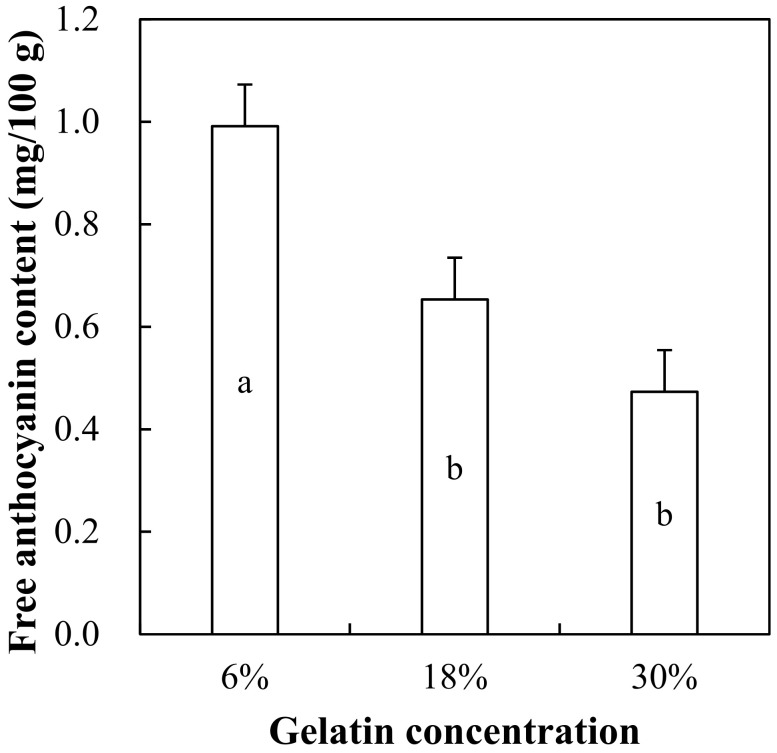
Free anthocyanin content of 6%, 18%, and 30% gelatin gels formulated with red maple leaf extracts (1000 mg/L). Error bars represent standard error of the means. a,b Means with the same letter are not significantly different (*p* ≥ 0.05).

**Figure 6 foods-10-03074-f006:**
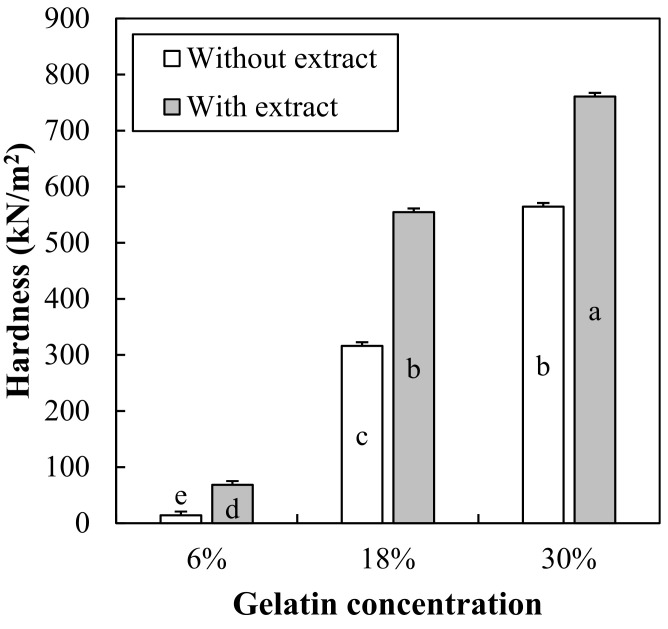
Hardness of 6%, 18%, and 30% gelatin gels formulated with red maple leaf extracts (1000 mg/L). Error bars represent standard error of the means. a–e Means with the same letter are not significantly different (*p* ≥ 0.05).

**Table 1 foods-10-03074-t001:** Color characteristics of 6%, 18%, and 30% gelatin gels formulated with red maple leaf extracts.

Traits	6% Gelatin Gel		18% Gelatin Gel		30% Gelatin Gels		SEM ^(1)^	Significance of *p* Value		
	without Extract	with Extract	without Extract	with Extract	without Extract	with Extract		G ^(2)^	E	G × E
L*	57.70 a	44.37 d	55.37 b	40.49 e	49.39 c	38.73 e	1.243	*** ^(3)^	***	*
(lightness)										
a*	0.17 e	10.40 c	0.34 de	11.41 b	0.42 d	12.09 a	0.933	***	***	***
(redness)										
b*	4.84 e	25.23 b	14.82 d	26.73 a	22.42 c	26.68 a	1.350	***	***	***
(yellowness)										
Hue angle	87.93 b	67.60 c	88.67 ab	66.88 c	88.92 a	65.60 d	1.849	NS	***	***

Treatment gel groups with extract were prepared with 1000 mg/L of hot water extract obtained from red maple leaf. ^(1)^ SEM: standard error of the mean. ^(2)^ G, gelatin concentration effect; E, red maple leaf extract effect; G × E, interaction effect. ^(3)^ NS, non-significance (*p* ≥ 0.05); *, *p* < 0.05; ***, *p* < 0.001. a–e Means with the same letter within a row are not significantly different (*p* ≥ 0.05).

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
