# Peer review of "Quality Characteristics of Senior-Friendly Gelatin Gels Formulated with Hot Water Extract from Red Maple Leaf as a Novel Anthocyanin Source"

_foods, 2021, doi:10.3390/foods10123074_

Round 1

Reviewer 1 Report

I thank the researchers for their efforts.

The following points must be considered.

Since you introduced the target group of this study to the elderly; and also mentioned that the consumption of foods containing antioxidants in the elderly can improve blood lipid concentrations and immune function; it is necessary to do these tests in the current research.

Unfortunately, the innovation of this research is not clear. It is obvious that as the concentration of gelatin increases, the firmness of the samples increases. It is also obvious that with increasing the concentration of anthocyanin extract, the color of the sample increases. No test has been performed to better understand some of the observed phenomena such as increased firmness with increasing extract concentration.

The thermal method for extracting heat-sensitive anthocyanin compounds is incorrect. It was better to use non-thermal treatments such as Pulsed electric field, microwave, and ultrasound to increase the extraction efficiency of the extract.

Line 89: It is mentioned that red maple leaves were dried for 3 hours at 50°C. It is necessary to mention the final moisture of the dried leaves. Had not anthocyanin compounds been thermally degraded at this temperature?

Line 91: Do not the solubility of anthocyanin compounds in acidic hydroalcoholic solvent have a higher extraction? Why use a water solvent at a neutral pH?

Line 92: I cannot find a justification for using high temperature (60°C) for anthocyanin compounds extraction. Please clearly state your justification for using this temperature.

Line 106: It is better to show the standard graph of Gallic acid and the constant value of k to determine total phenolic content.

Line 142: Why did you use the 120 minute sample for this stage of the experiments?

Line 163: Why is the diameter of the probe used in this test less than the diameter of the square surface of your sample? Can a compression test be performed with this probe diameter?

Results section: The verbs used in the results section should be used in the past tense. Please follow the whole of the manuscript.

Section 3.1: Why did the total phenolic content of the samples not change with increasing extraction time? Please provide a proper justification. It is also necessary to make good comparisons with similar articles.

Line 198 and 228: Please specify meaningful letters in Figures 1 and 2.

Performing dynamic rheological tests, scanning electron microscope of the gel samples produced is necessary to provide a good justification for the observations.

Analysis of anthocyanin compounds of this new source is extremely necessary.

In the conclusion, the specific and bold results of the research should be mentioned. Therefore, this section needs a serious rewrite.

Author Response

Thank you for your comments. Please see attached file to check revised parts.

Reviewer 2 Report

In the manuscript by Song et al entitled “Hot water extract from red maple leaf as a novel source of anthocyanin in senior-friendly gelatin gels” the authors evaluate the antioxidant activity of red maple leaf hot water extract and their impacts on colour, free anthocyanin content, and hardness of gelatin gels. The potential use of red maple leaf extract could enhance antioxidant content in senior-friendly jelly food although the limited amount. Overall, the manuscript is well written, the results are well performed and the conclusions are convincing. I suggest some minor revisions.

Page 1, line 13: change ‘ 60ºC’ by ’60 °C’. Check this in overall text;

Page 1, lines 31-33: not only plants in general but also seeds contain antioxidant compounds: see for examples legumes as lentils or beans: Landi et al, Food Funct. 2015 Sep;6(9):3155-64; Landi et al, Acta Sci Pol Technol Aliment. 2017 Jul-Sep;16(3):331-344;

Page 3, line 100: change ‘One hundred milligrams’ by ‘100 mg’; moreover, in this paragraph add the calibration range;

Page 3, line 110: change ‘0.025M’ by ’25 mM’;

Page 3, lines 116-117: add unit of measure for cyanidin-3-glucoside and molar absorptivity;

Figure 1. use only the letter for the two graph (adding ‘a’ or ‘b’ on top-left of each graph) since the figure legend already describe the meaning of ‘a’ and ‘b’. The same for Figures 2 and 3. Moreover, in Figure 1, add unit of measure also for GAE (e. g. mg GAE g-1) in the title of y axis of ‘graph a’.

Author Response

(The authors gave the same response as above.)

Reviewer 3 Report

Recommendation: minor revision

Manuscript number: foods-1500858-peer-review-v1

Article Type: Article

Title: Hot water extract from red maple leaf as a novel source of anthocyanin in senior-friendly gelatin gels

Comments:

Lines 40-41: Maple (Acer palmatum Thunb.) is naturally grown in East Asia, North America, and 40 Europe, and 20% of its total compounds are composed of flavonoid compounds [12].

Here is two times word “compounds” in the sentence

Lines 42-45: too long sentence

Table 1: I recommend put in the firs column, only symbols of parameters, e.g. L*, a* b*, without CIE

Figure 6: I recommend change the unit from N/m2 to kN/m2

Author Response

(The authors gave the same response as above.)

Round 2

Reviewer 1 Report

Thank you for your effort.
But unfortunately, lots of hints that have been said have not yet been corrected.
Various tests are needed to understand the content and findings of your research that you said you could not do.

Author Response

Thank you for your comments again. Moreover, we are fully agreement with your suggestion that more scientific assay should be conducted to show the characteristics of red maple leaf extract as an anthocyanin source. However, this study have been tried to show the addition effect of hot water extract of red maple leaf on general quality of gelatin gels. In this regard, we have made major two changes to solve your concern as well as to exactly present our research purpose.

1. The title of manuscript was changed as follows; 

Quality characteristics of senior-friendly gelatin gels formulated with hot water extract from red maple leaf as a novel anthocyanin source

2. We have mentioned the necessity of further studies showing the functionality of red maple leaf extract as a novel anthocyanin source, which has been additionally described in the conclusion. 

Furthermore, in order to evaluate the effect of the hot water extract of maple leaf on the functionality of the senior-friendly gels as an anthocyanin source, further studies profiling the anthocyanin composition of red maple leaf extract as well as developing an efficient extraction should be warranted.

We hope that our revision may be sufficient to solve your concerns.

Thanks again.